# Fatigue Life Evaluation Considering Fatigue Reliability and Fatigue Crack for FV520B-I in VHCF Regime Based on Fracture Mechanics

**Jinlong Wang [1],\*, Yuxing Yang [1], Jing Yu [1], Jingsi Wang [1], Fengming Du [1] and Yuanliang Zhang [2]**

[1]   Marine engineering college, Dalian Maritime University, Dalian 116000, China;
    yangyuxing@dlmu.edu.cn (Y.Y.); yj_lunji@dlmu.edu.cn (J.Y.); wjs@dlmu.edu.cn (J.W.);
    dfm@dlmu.edu.cn (F.D.)
[2]   School of mechanical engineering, Dalian University of Technology, Dalian 116000, China;
    zylgzh@dlut.edu.cn
\*   Correspondence: wjl19890806@dlmu.edu.cn; Tel.: +86-150-4111-8408

**Abstract:** This paper focuses on the fatigue reliability analysis and the development of a new life model of reliability and crack growth mechanisms in FV520B-I (high strength martensitic-type stainless steels) in the very-high cycle fatigue (VHCF) regime, which haven't been studied well. First, the fatigue test was carried out to clarify the fatigue failure mechanism in the very-high cycle regime. Based on the test results and fatigue reliability theory, the fatigue life distribution and *P-S-N* curves were modeled. A new fatigue life evaluation model for FV520B-I is proposed according to the fracture mechanics and classic life evaluation method. With the comprehensive application of *P-S-N* curves and a new proposed fatigue life evaluation model, a new assumption of a *P-Sc-N* curve is developed and verified, to quantitatively express the relationship between fatigue life, reliability and fatigue cracking. This is novel and valuable for further fatigue study of FV520B-I.

**Keywords:** FV520B-I; fatigue reliability; *P-Sc-N* curve

## 1. Introduction

FV520B-I, as an important high-strength engineering metal, has been widely adopted in the manufacturing of centrifugal compressor vanes for its positive mechanical properties, including high strength, high corrosion resistance, high abrasive resistance and good welding characteristics [1,2]. In practical engineering, the fatigue life of the FV520B-I component is required to be more than $10^7$ cycles, a very-high cycle fatigue (VHCF) level. Quantitatively expressing the fatigue reliability of FV520B-I is a key research topic, especially for those who can use the information to improve the manufacturing of mechanical equipment and avoid fatigue failure. Fatigue reliability [3–8] refers to the probability that theoretical fatigue life will meet the actual required life time of the material, which can be quantitatively expressed by a probability of the fatigue life without the fatigue failure in solving real engineering fatigue problems, especially for the core components of large mechanical equipment. With FV520B-I being more and more widely used, there is increasing interest in detailed study modelling its fatigue reliability in the VHCF regime, and developing a new precise fatigue life evaluation model (which is beneficial to clarify fatigue failure characteristics, and also very essential to avoid fatigue failure efficiently in real engineering practice).

A few investigations have been carried out to explore the fatigue reliability of metallic materials. Wirsching [9–12] studied probabilistic and statistical methods for assessing fatigue in design and analysis. New methods were proposed to analyze the reliability of both individual joints and the overall system, with consideration of crack initiation. Wang [13] conducted a study of the fatigue

mechanisms of conveyor slide rails and wheels under constant amplitude cyclic loading, and a normal distribution model was employed to make a contrastive analysis between the fatigue reliabilities of rails and wheels. Wang [14] conducted a bending fatigue experiment on metal 8822H to establish the *P-S-N* curve and analyze fatigue reliability of metal 8822H. Wurzel [15] focused on the fatigue of aerospace components, and a new fatigue evaluation model was proposed based on the classical theory and experimental data. Paolino [16] put forward an investigation about the effect of fatigue cracks on the *P-S-N* curve for metallic material in the VHCF regime.

For predicting the fatigue life of FV520B-I, some previous studies have investigated the relationship between fatigue cracks and fatigue life. Liu [17,18] performed a finite element analysis for the open impeller to clarify the stress distribution of FV520B-I impeller in actual operational conditions, and performed a fatigue analysis based on classic fatigue theory. Zhang [19–22] focused on a study of the effect of fatigue influence parameters (including internal inclusion, surface roughness and loading frequency) on the fatigue failure mechanism of FV520B-I in VCHF and HCF regime. Also, the quantitative expression model between the influence parameters and fatigue life were proposed and verified.

Some research has been carried out on fatigue reliability analysis and fatigue properties of FV520B-I, and these studies provide a theoretical foundation for the development of fatigue reliability analysis methods and establishment of FV520B-I fatigue life equation. However, up to now, there is very little scientific understanding of the fatigue reliability of FV520B-I in the VHCF regime. No previous specific study of FV520B-I in the VHCF regime has been performed to address the question of *P-S-N* curve establishment of FV520B-I, and neither has a fatigue life evaluation model considering reliability and fatigue cracks in FV520B-I been well developed. More than that, fatigue reliability analysis and life evaluation for FV520B-I in engineering practice is very case-oriented—different theories and models are specific to desired applications and conditions, and the parameters of the model are not the same when applying to dissimilar material modeling.

The study presented here is one of the first specific investigations to explore the fatigue reliability of FV520B-I in the VHCF regime to address the questions above. The primary aim of this paper is to contribute to the quantitative expression of fatigue reliability for FV520B-I in the VHCF regime, and attempt to develop a new fatigue life evaluation model combining reliability and fatigue cracking together. As shown in Figure 1. First, an ultrasonic fatigue test for FV520B-I was carried out to collect effective fatigue test data for FV520B-I and provide scientific support for the following fatigue reliability estimation. The characteristics of the fracture surface were observed with SEM to make clear the leading cause of fatigue failure. Next, the fatigue life distributions of FV520B-I in the VHCF regime and *P-S-N* curve models with different reliability values were established, with the comprehensive application of a three-parameter model, test data and a logarithmic normal distribution function. The classic fatigue life formula was employed to develop a new fatigue life evaluation model for FV520B-I with the consideration of fatigue reliability, and the new proposed model was verified to be reliable through additional fatigue tests. This demonstrates that the new model can provide satisfactory reliability analysis results and life predictions for FV520B-I. Lastly, an assumption of a *P-Sc-N* curve is proposed which combines fatigue reliability, fatigue cracking, and fatigue life together, and the *P-Sc-N* curve is modeled with three different reliability values. Further study is still necessary to demonstrate its rationality and correctness. A framework has been created in establishing the analytical *P-S-N* and life evaluation models for FV520B-I in the VHCF regime, and this will enhance the accuracy of the fatigue reliability analysis in real engineering practice. This study is valuable to engineering practice and very supportive to the theoretical study of fatigue reliability.

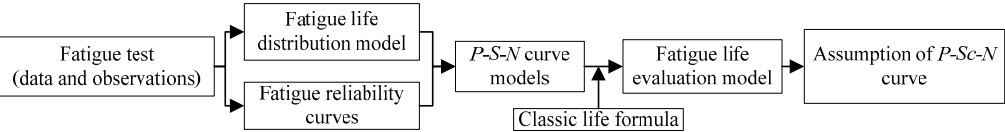

**Figure 1.** The structure of the research on proposed fatigue reliability.

## 2. Materials and Methods Fatigue Test

### 2.1. Test Method

As mentioned above, lack of fatigue data has existed as a primary problem in research on the fatigue properties of FV520B-I. Fatigue testing is the most common and effective method to obtain useful data and provide practical support for theoretical fatigue analysis. In order to investigate the fatigue reliability of FV520B-I in VHCF regime, a fatigue test was carried out in a USF-2000 ultrasonic fatigue test system. This fatigue test system consists of a control and data acquisition part, a monitoring system, and a test implementation part, as shown in Figure 2a. The operating frequency was set to 20 KHz. The maximum stress amplitude used in the test was 600 MPa and the minimum was 500 MPa. The interval was 25 MPa. The mean stress in the test was set to 0, that is, with stress ratio $r = -1$ (symmetric cyclic load) and the stress amplitude means the alternating stress from 0. Air cooling and a pulse-pause loading model are employed to avoid the effect of temperature rise on the test results.

### 2.2. Material and Specimen

The test object was high-strength metal FV520B-I; the chemical composition of FV520B-I is shown in Table 1. The mechanical properties of FV520B-I are shown in Table 2.

**Table 1.** Chemical composition of FV520B-I (wt%).

| Chemical Composition | C | Si | Ni | Cu | S | Cr | Mo | Nb | Fe |
|---|---|---|---|---|---|---|---|---|---|
| Content | 0.02–0.07 | 0.15–0.7 | 5–6 | 1.3–1.8 | <0.025 | 13–14.5 | 1.3–1.8 | 0.25–0.45 | Bal |

**Table 2.** Mechanical property of FV520B-I.

| Mechanical Parameters | Elastic Modulus $E$ (GP) | Tensile Strength $R_m$ (MPa) | Yield Strength $R_{p0.2}$ (MPa) | Vickers Hardness HV | Elongation A (%) |
|---|---|---|---|---|---|
| FV520B-I | 194 | 1180 | 1029 | 380 | 16.07 |

A standard specimen was employed in the test. The geometry of the specimen is shown in Figure 2b, and the units in Figure 2b are mm.

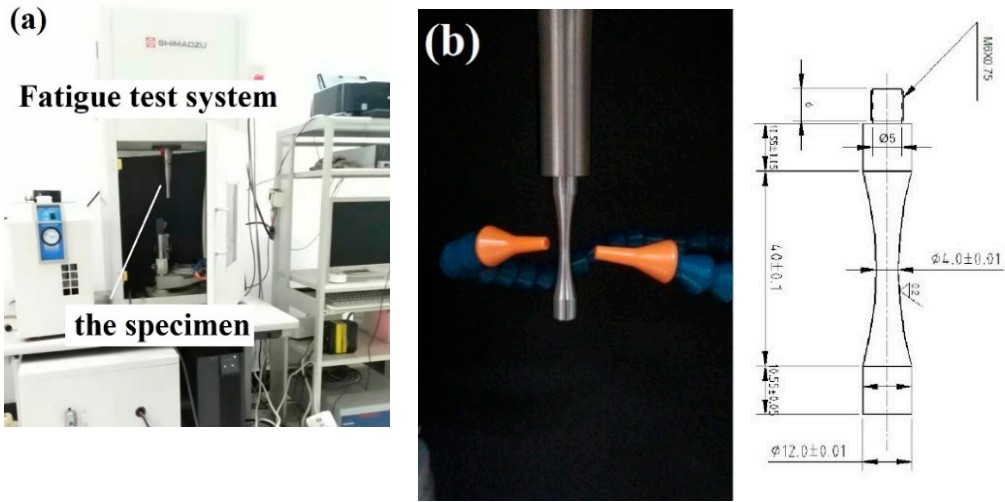

**Figure 2.** The fatigue test system and specimen. (**a**) The fatigue test system. (**b**) The specimen and its dimensions.

### 2.3. The Results and Observations

The test results include the stress amplitude and fatigue life, as shown in Figure 3.

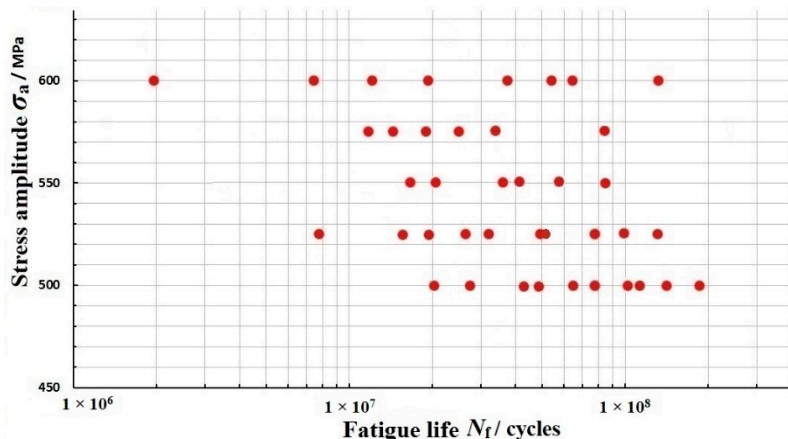

**Figure 3.** The test results for fatigue.

Several fatigue test phenomena of FV520B-I have been observed with the application of scanning electron microscopy (SEM), such as Granular Bright Facet (GBF) (JEOL Ltd, Tokyo, Japan) and "fish-eye". Shiozawa [23–25] and Murakami [26–28] focused on the study of VHCF failure mechanisms. It was proposed that primary crack initiation sites are around the subsurface crack: during the fatigue process, some cracks form in the GBF and continue propagating to "fish-eye" until failure. These two fracture features are prominent characteristics of VHCF failures of metallic materials.

The GBF region is captured clearly through SEM observation of the specimen in the experiment as shown in Figure 4. This is the most significant characteristic of the fatigue failure caused by subsurface crack (or internal inclusion).

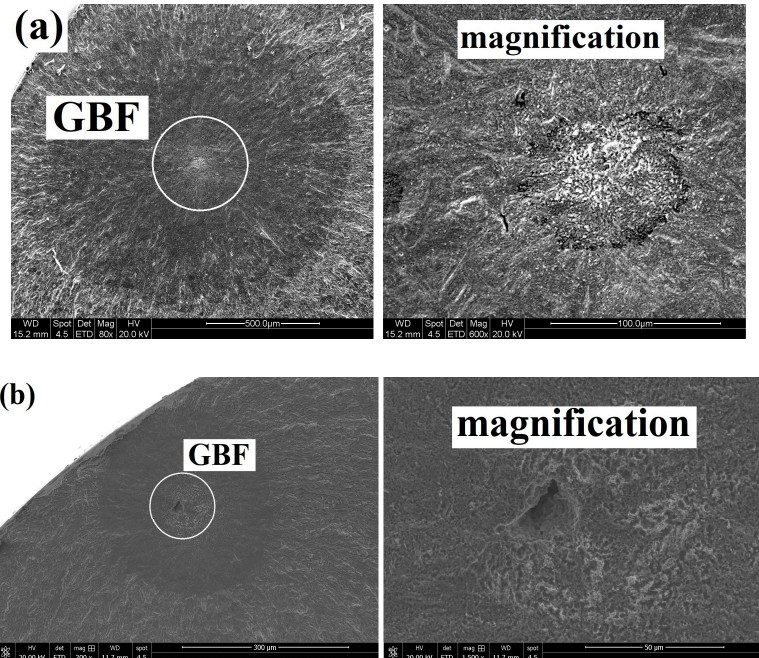

**Figure 4.** The GBF (Granular Bright Facet) region. (**a**) the GBF region, (**b**) the GBF region.

Except in the GBF region, from the fracture surface of the specimen, the "fish-eye" boundary can be clearly observed as shown in Figure 5. "Fish-eye" is the most significant characteristic of very-high

cycle fatigue fracture, and the change of the crack propagation rate during the crack propagation process leads to the formation of "fish-eye."

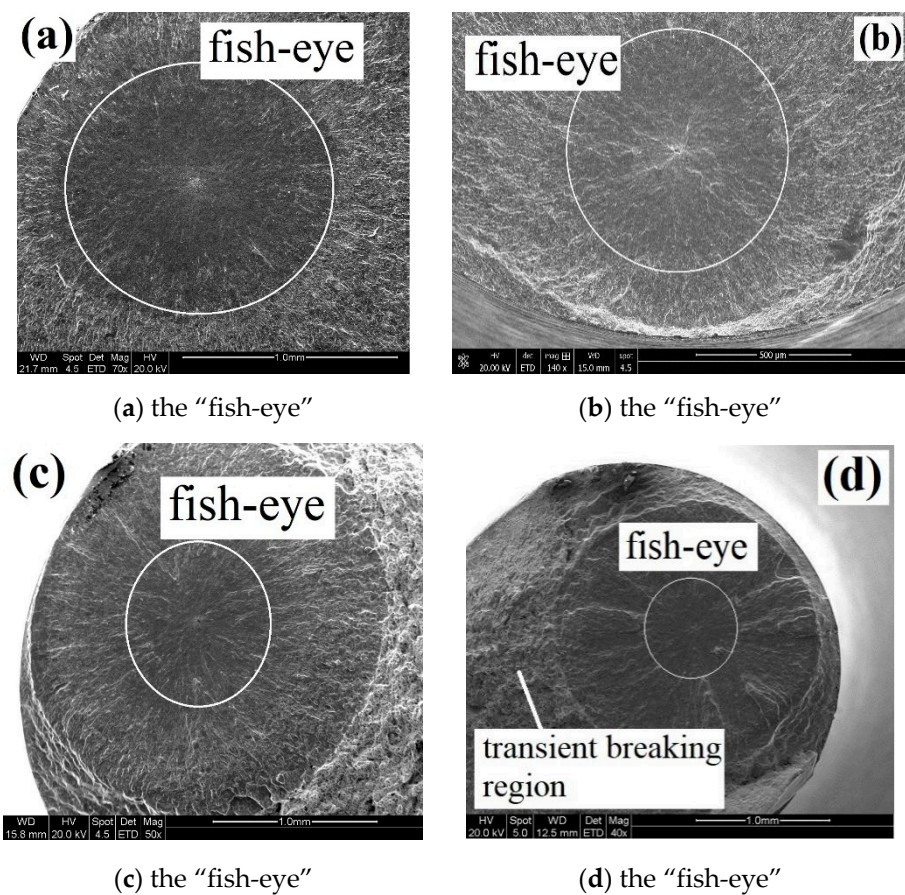

(**a**) the "fish-eye"　　　　　　　　　　　　(**b**) the "fish-eye"

(**c**) the "fish-eye"　　　　　　　　　　　　(**d**) the "fish-eye"

**Figure 5.** The "fish-eye" region.

## 3. Fatigue Reliability of FV520B-I in the VHCF Regime

### 3.1. Very-High Cycle Fatigue Life Distribution of FV520B-I in the VHCF Regime

Based on the test results, as shown in Figure 2, the fatigue life distribution of FV520B-I in the VHCF regime with different stress amplitudes can be estimated. The logarithmic normal distribution model has been widely used to quantitatively analyze the reliability of various mechanical parts for its numerous advantages: a simple calculation process, high accuracy, and wide applicability. Logarithmic normal distribution is employed as the base model in this paper to obtain the fatigue life distribution of FV520B-I in the VHCF regime. The fatigue life probability density function [29] of the logarithmic normal distribution is written as:

$$f(x) = \frac{1}{x\sigma\sqrt{2\pi}}exp\left[-\frac{1}{2}\left(\frac{\ln x - \mu}{\sigma}\right)^2\right] \tag{1}$$

where $x$ is the actual fatigue life in the engineering condition, $N_f$; $\mu$ is location parameter, and $\sigma$ is proportional parameter. $\mu$ and $\sigma$ are expressed as:

$$\mu = \frac{\sum_{i=1}^{n} ln x_i}{n} \tag{2}$$

$$\sigma = \sqrt{\frac{\sum_{i=1}^{n}(lnx_i - \mu)^2}{n-1}} \tag{3}$$

In order to guarantee the accuracy of the life distribution model, appropriate data must be selected from the test to estimate $\mu$ and $\sigma$. $x_i$ is the $i$th variable; here, it is the fatigue life of FV520B-I with a certain stress amplitude as shown in Figure 3. As displayed in Equation (2), the model parameter $\mu$ means the average of logarithm of fatigue life, and the model parameter $\sigma$ is the standard deviation of the logarithm of fatigue life.

The *P-S-N* (*P* is reliability, *S* is stress and *N* is fatigue life) curve is an important and practical method in engineering conditions, especially for those who can use *S-N* curves to quickly determine the fatigue reliability and life, to improve the reliability of the FV520B-I component design and avoid fatigue failure for FV520B-I. Thus, it is a key issue to address the problem of lacking *P-S-N* curves for FV520B-I, and establish a *P-S-N* curve model to quantitatively analyze the fatigue reliability of FV520B-I in the VHCF regime. According to the definition of the *P-S-N* curve, the *P-S-N* curve actually means the *S-N* curve with a certain reliability value *R*. Thus, determining the *S-N* curve model is the primary work in *P-S-N* curve modeling. The commonly used *S-N* basics formula [30] is written as Equation (4):

$$\text{Three} - \text{parameter model}: \ (S - S_0)^{\alpha}N_{\text{f}} = C \tag{4}$$

where *S* is the stress amplitude, which is equal to the $\sigma_a$ in the fatigue test; $N_{\text{f}}$ is fatigue life; $S_0$ is the stress parameter, and $\alpha$ and *C* are constant parameters. These three parameters are unknown for FV520B-I. A three-parameter model [29–31] was chosen as the basic model to deduce the *P-S-N* curve of FV520B-I in this paper, for its two advantages compared to other models: flexible form and better ability of data fitting.

Furthermore, at least three groups of test data with different stress amplitudes are required, from 500 MPa to 600 MPa, as employed in the fatigue test. Adequate amounts of test data are required to ensure the accuracy of estimation results. The fatigue results in Figure 3 include the fatigue life with five different stress amplitude values; most of the test data are in the very-high cycle fatigue regime, which means the test data are sufficient for the estimation of the unknown model parameters.

In order to realize the simple calculation process and obtain accurate calculation results, the fatigue test data with three stress amplitude 500 MPa, 525 MPa and 600 MPa were employed with a comprehensive consideration of the distribution range of test data and the amount of test data. By employing the test data in the very-high cycle fatigue regime in Figure 3, the three groups of the unknown parameter $\mu$ and $\sigma$ are estimated; the results are shown in Table 3.

**Table 3.** The parameters $\mu$ and $\sigma$.

| Stress Amplitude | 500 MPa | 525 MPa | 600 MPa |
|---|---|---|---|
| location parameter $\mu$ | 2.1673 | 1.6429 | 1.0239 |
| proportional parameter $\sigma$ | 0.4372 | 0.4213 | 0.8989 |

By substituting the parameters in Table 3 into Equation (1), the probability density functions of the logarithmic normal distribution under different stress amplitudes are obtained, and the corresponding probability density curves are shown in Figure 6.

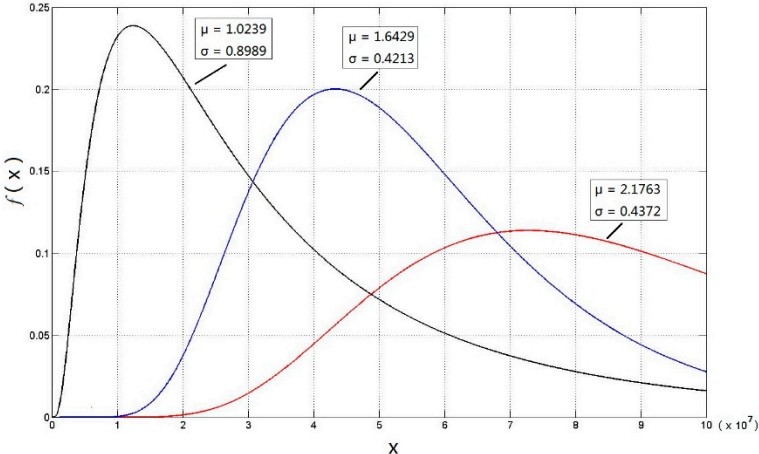

**Figure 6.** The fatigue life probability density function curves of FV520B-I.

The reliability of the logarithmic normal distribution function can be expressed as:

$$R(X) = P(X > x) = 1 - \Phi\left(\frac{\ln x - \mu}{\sigma}\right), \ (x > 0) \tag{5}$$

where $x$ is $N_f'$, which presents the actual fatigue life in the engineering condition (it conforms to the fatigue life distribution); $X$ is $N_{fT}$, the required fatigue life without failure; and $R$ is the fatigue reliability, which presents the probability with $N_{fT}$ as the variable when the actual fatigue life has reached or exceeded the required life without fatigue failure (that is the probability of $N_{fT} > N_f'$). In a word, the reliability is always defined as the probability that, for a given stress amplitude and defect size, the random variable fatigue life, $N_{fT}$, is larger than the required fatigue life, $N_f'$: $R = P(N_{fT} > N_f')$.

Fatigue reliability curves based on the logarithmic normal distribution model are obtained according to Equation (5), as shown in Figure 7. A negative correlation between fatigue reliability and fatigue life can be clearly detected: with the increased fatigue life, reliability without failure decreases.

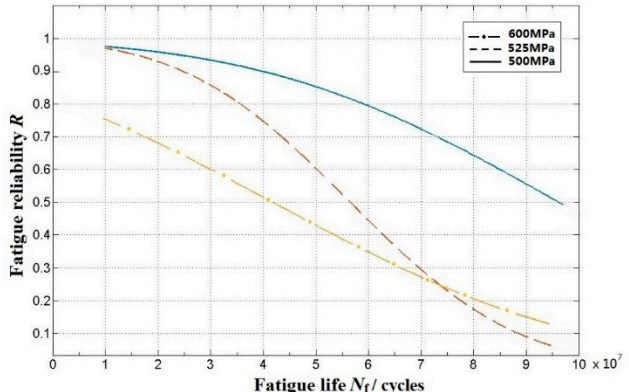

**Figure 7.** The fatigue reliability curves.

### 3.2. The P-S-N Curve Estimation

As discussed above, a three-parameter model was employed to estimate the *P-S-N* curve of FV520B-I, which has not been published before. The three-parameter model [32,33] (Equation (6)) is the extension and improvement of the Basquin model, and is proposed for estimating *P-S-N* curves to describe the slope part and horizontal part. With an expansion of application scope, the three-parameter model was also employed to estimate *P-S-N* curves in VHCF [34]. The *P-S-N* curve refers to the *S-N* curve corresponding to different survival rates *P* (*P* = 1 − *R*), considering the dispersion of fatigue life.

Each certain fatigue reliability $R$ corresponds to a certain *P-S-N* curve, so the three-parameter model with a certain fatigue reliability $R$ can be re-expressed as:

$$(S - S_{0,R})^{\alpha_R} N_f = C_R \tag{6}$$

where $S_{0,R}$ is the stress parameter when the reliability is $R$, and $\alpha_R$ and $C_R$ are the unknown material parameters with fatigue reliability $R$. At least three *S-N* curves with different fatigue reliability values should be modeled to constitute the *P-S-N* curves of FV520B-I.

In most situations, the generic *S-N* curve is the *P-S-N* curve with reliability $R = 0.5$, and is also the most widely used *P-S-N* curve in practical applications to analyze the fatigue reliability and fatigue life of FV520B-I in VHCF regime. Thus, the *P-S-N* curve with $R = 0.5$ is chosen as an example to show the deduction procedure of *P-S-N* curves with different fatigue reliability values. As the three-parameter model has been widely accepted as a way to obtain the *P-S-N* curve, so the derivation procedure of the *P-S-N* curve for FV520B-I is straightforward. The basic three-parameter model with $R = 0.5$ under different stress amplitude values can be written as:

$$\begin{cases} (S_1 - S_{0,0.5})^{\alpha_{0.5}} N_{f0.5S_1} = C_{0.5} \\ (S_2 - S_{0,0.5})^{\alpha_{0.5}} N_{f0.5S_2} = C_{0.5} \\ (S_3 - S_{0,0.5})^{\alpha_{0.5}} N_{f0.5S_2} = C_{0.5} \end{cases} \tag{7}$$

where $S_{0,0.5}$, $\alpha_{0.5}$ and $C_{0.5}$ are unknown model parameters for $R = 0.5$; $S_1$, $S_2$, and $S_3$ are three stress amplitude values: $S_1 = 500$ MPa, $S_2 = 525$ MPa and $S_3 = 600$ MPa; and $N_{0.5,S1}$, $N_{0.5,S2}$ and $N_{0.5,S3}$ are fatigue life with $R = 0.5$ under different stress amplitudes, which can be calculated with the combined application of the standard normal distribution table and Equation (5).

Next, the fatigue life values with different stress amplitude values are calculated as expressed in Equation (8):

$$\begin{cases} N_{0.5S_1} = \exp(\mu_1) = \exp(1.0239) = 2.78(\times 10^7) \\ N_{0.5S_2} = \exp(\mu_2) = \exp(1.6429) = 5.17(\times 10^7) \\ N_{0.5S_3} = \exp(\mu_3) = \exp(2.1673) = 8.73(\times 10^7) \end{cases} \tag{8}$$

Moreover, the model data that are employed to obtain the *P-S-N* curve with $R = 0.5$ are shown in Table 4.

**Table 4.** Stress amplitude, fatigue reliability, and fatigue life.

| Stress Amplitude $\sigma_a$/MPa | $S_1$ | $S_2$ | $S_3$ |
|---|---|---|---|
| The fatigue reliability $R$ | 0.5 | 0.5 | 0.5 |
| Fatigue life $N_f$ | $2.78 \times 10^7$ | $5.17 \times 10^7$ | $8.73 \times 10^7$ |

This basic model (Equation (7)) is established based on classical fracture mechanics and mathematical derivations, so the suitability of the model for the high-cycle fatigue life prediction of FV520B-I is assured in nature. Each individual material will have its own model with particular parameters, but there are three unknown model parameters for FV520B-I, including $S_{0,0.5}$, $\alpha_{0.5}$ and $C_{0.5}$. These parameters are dependent on metal FV520B-I and no shared values can be adopted for the model identification directly. These three parameters are essential in the establishment of the life model. A small change in them will result in a significant alteration in the fatigue life prediction results. This implies that there is no published model that can cover a range of different materials. Thus, it is key to identify the unknown parameters to develop a specific model for FV520B-I with $R = 0.5$ in VHCF regime.

As shown in Table 4, the data are diverse, not following certain rules. A nonlinear fitting algorithm should be adopted to identify the unknown parameters. Such a fitting algorithm uses continuous curves or analytical expressions to represent discrete data and the functional relationship between the

variables. The underdeveloped base model is established in Equation (9), and iterative computation is employed to determine the value of $S_{0,0.5}$, $\alpha_{0.5}$ and $C_{0.5}$ for the function $f(S_i, N_{fi})$ using the minimizing operation of Equation (9), i.e.,

$$(S_{0,0.5}, \alpha_{0.5}, C_{0.5}) = \underset{S_{0,0.5}, \alpha_{0.5}, C_{0.5}}{\text{argmin}} \sum_{i=1}^{3} (f(Sc_i, N_{fi}) - N_i)^2 \tag{9}$$

With employing the model data in Table 4 into Equation (6), the unknown model parameters are estimated as shown in Table 5.

**Table 5.** Model parameters.

| $S_{0,0.5}$ | $\alpha_{0.5}$ | $C_{0.5}$ |
|:---:|:---:|:---:|
| 481.1823 | 0.6205 | 53.9676 |

By substituting the model parameters into Equation (7), the *S-N* curve model of FV520B-I with $R = 0.5$ is developed as:

$$(S - 481.1823)^{0.6205} N = 53.9676 \tag{10}$$

A *P-S-N* curve with $R = 0.5$ has been determined in Equation (10), and more *P-S-N* curves with different fatigue reliability values should be developed to enrich the *P-S-N* curves of FV520B-I. Two more *P-S-N* curves in critical conditions with $R = 0.9$ and $R = 0.1$ are employed as additional examples. When the reliability $R = 0.9$, it means that there is a high probability that the fatigue life of FV520B-I can satisfy the requirement. Conversely, when the reliability $R = 0.1$, the fatigue life of FV520B-I likely can't satisfy the requirement. According to the same deduction process displayed above, *S-N* curves with $R = 0.9$ and $R = 0.1$ are modeled, and three-parameter models with three different fatigue reliability values (0.1, 0.5 and 0.9) are expressed below, respectively:

$$\begin{cases} R = 0.1: (S - 499.2021)^{0.0715} N_f = 11.5501 \\ R = 0.5: (S - 481.1823)^{0.6205} N_f = 53.9676 \\ R = 0.9: (S - 478.0581)^{1.1550} N_f = 226.8041 \end{cases} \tag{11}$$

Based on Equation (11), three *P-S-N* curves for FV520B-I in VHCF regime can be drawn as shown in Figure 8.

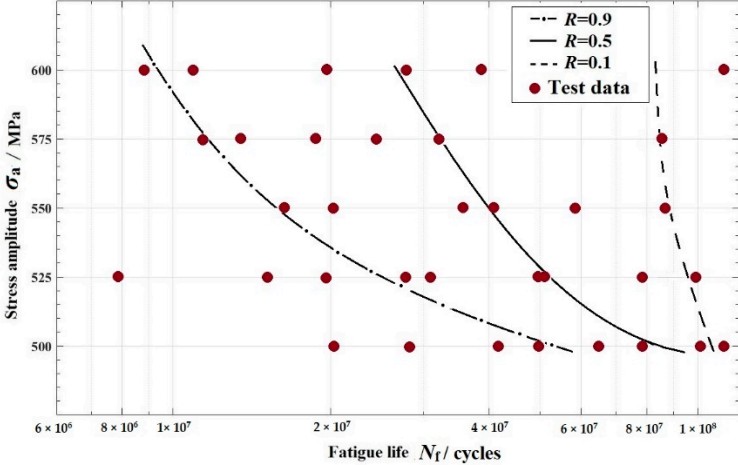

**Figure 8.** *P-S-N* curves with the $R = 0.1, 0.5, 0.9$.

### 3.3. Discussion

Based on Figure 8 and Equation (11), the combination $(R, N_f, \sigma_{ar})$ can be obtained directly, and this has important practical application value for actual engineering. Two characteristics of the FV520B-I fatigue reliability can be figured out according to the *P-S-N* curves. First, in the case of constant stress amplitude, there is obvious negative correlation between the reliability and fatigue life of FV520B-I: with increased fatigue life, fatigue reliability decreases. For example, with $\sigma_{ar} = 575$ MPa, when the fatigue life is around $10^7$ cycles, the fatigue reliability is about 0.9. This means the fatigue life of FV520B-I can achieve the VHCF level with a high probability. However, when the fatigue life increases to about $10^8$ cycles, fatigue reliability decreases from 0.9 to 0.5, even as low as 0.1, which means that the fatigue failure can easily occur in the regime at $10^8$ cycles.

Second, Figure 8 indicates that the decreasing the stress amplitude is an effective way to increase fatigue life and enhance fatigue reliability at the same time. It should be noticed that the stress amplitude used in the fatigue test is the maximum stress; this is calculated using the applied fatigue load set by the test system and the interface size of the specimen. However, in practical engineering conditions, the influence of stress amplitude depends on the mean stress and, particularly, on fatigue crack size. Under the same applied loading conditions, the specimen with smaller fatigue cracks will suffer smaller stress amplitudes, which will result in long fatigue life and high reliability. Meanwhile, large fatigue cracks will lead to serious stress concentration and big stress amplitude on the FV520B-I component, so fatigue life will be reduced and failure will easily occur.

So, an assumption is proposed based on the above theoretical analysis: that the influence of stress amplitude on fatigue reliability can be replaced by the influence of fatigue cracks on fatigue reliability; there should be a quantitative relationship between reliability, fatigue cracking, and fatigue life.

## 4. The Fatigue Life Evaluation for FV520B-I

### 4.1. The Fatigue Life Evaluation Model Considering Reliability

It has been reported and demonstrated that the fatigue cracking is the leading cause of VHCF failure of FV520B-I; the fatigue test results in this paper are consistent with the reported conclusions. Up to now, the reported fatigue life prediction model for FV520B-I can only quantitatively describe the single relationship between fatigue cracking and fatigue life, and the relevant fatigue reliability in this situation is not clear. In order to provide more comprehensive theoretical support for the maintenance of FV520B-I specimens in practical engineering, a new fatigue life evaluation model for FV520B-I in the VHCF regime is required to quantitatively express fatigue reliability and life. This new model is creatively developed with the combined application of the reported life model and a *P-S-N* curve, which can be employed to evaluate reliability and life at the same time, to improve the applicability of FV520B-I fatigue analysis results in practical engineering.

The fatigue life of FV520B-I in VHCF regime is employed as the transition variable of to integrate the reliability and crack into one united model. Thus, the first step in the development of this new model is to determine the fatigue life of FV520B-I. Zhang [20] proposed an specific very-high cycle fatigue life model for FV520B-I, as displayed in Equation (12)

$$N_f = \frac{9\Delta K_{th}^2 G}{2E(\sigma_a - \sigma_r)^2 Sc} + \frac{Sc^{(1-n/2)}}{C\sigma_a{}^n \beta_1^n \pi^{n/2}\left(\frac{n}{2} - 1\right)} \tag{12}$$

where $Sc$ is the fatigue cracking in μm (here the inclusion size for FV520B-I); HV is Vickers hardness, HV = 380, kgf/mm$^2$; $\sigma_a$ is the stress amplitude, MPa; elasticity modulus $E = 2.1 \times 10^5$, MPa; shear modulus $G = 8 \times 10^4$, MPa; $\beta_1$ is the geometric constant $\beta_1 = 0.5 \times \pi^{0.5}$; $\Delta K_{th}$, $C$, $n$ and $\alpha$ are the material parameters: $\Delta K_{th} = 6.3$ MPa·m$^{1/2}$, and $C$ and $n$ are dimensionless constants: $C = 3.14 \times 10^{-14}$, $n = 2.95$ [20].

In order to keep the same stress amplitude conditions in the derivation process of the *P-S-N* curve model of FV520B-I, stress amplitudes of 600 MPa, 525 MPa and 500 MPa were employed. The quantitative relationship curves between fatigue crack and fatigue life with 600 MPa, 525 MPa and 500 MPa were drawn in Figure 9, based on Equation (12).

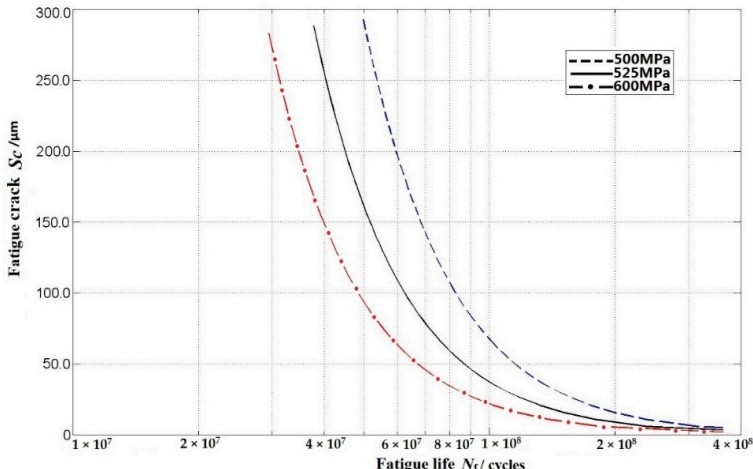

**Figure 9.** Fatigue life curves with 500 MPa, 525 MPa, and 600 MPa.

The second step is to combine the fatigue reliability, fatigue life and cracking into one unified model. As mentioned above, in most situations, the *P-S-N* curve with reliability *R* = 0.5 has been widely accepted as the generic *S-N* curve, for investigating or quantitatively expressing the fatigue properties of metallic material. Thus, *R* = 0.5 is also chosen as the basic situation in the new fatigue life evaluation model of FV520B-I in the VHCF regime. The fatigue life with *R* = 0.5 under three different stress amplitude values is estimated as shown in Table 4. By substituting the fatigue life with *R* = 0.5 into Equation (12), the corresponding fatigue cracking can be obtained as displayed in Table 6 and Figure 10.

**Table 6.** Stress amplitude, fatigue reliability, fatigue life and fatigue cracking.

| Stress Amplitude $\sigma_a$/MPa | 600 | 525 | 500 |
| --- | --- | --- | --- |
| Fatigue reliability *R* | 0.5 | 0.5 | 0.5 |
| Fatigue life $N_f$ | $2.78 \times 10^7$ | $5.17 \times 10^7$ | $8.73 \times 10^7$ |
| Fatigue crack *Sc*/μm | 322 | 139.4 | 90.6 |

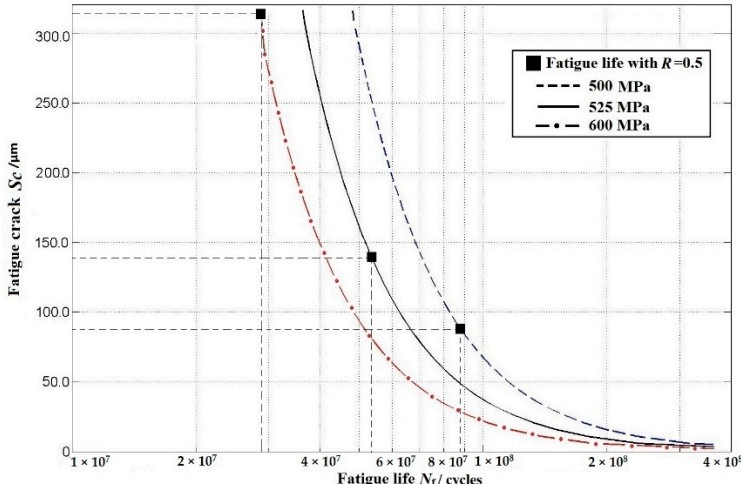

**Figure 10.** The fatigue life with *R* = 0.5.

It is clear that there is a non-linear relationship between fatigue cracking (*Sc*) and fatigue life ($N_f$) with $R = 0.5$. However, no previous model has been reported to fit this relationship based on the distribution characteristics of fatigue life with $R = 0.5$ that is a nonlinear exponential distribution. Thus, the form of the two-parameter exponential model is employed as a reference to develop the new fatigue life evaluation model. This model is employed to develop the quantitative expression between reliability, fatigue cracking, and fatigue life, as displayed:

$$Sc^{a_R} N_f = b_R \tag{13}$$

where *Sc* is fatigue cracking, μm; $N_f$ is fatigue life; and $a_R$ and $b_R$ are model parameters related to the fatigue reliability of FV520B-I. With the same derivation process as Equation (10) and the data in Tab.6, a nonlinear fitting algorithm should be adopted to identify the unknown parameters in Equation (13). The estimated results for the model parameters are: $a_{0.5} = 0.91$, $b_R = 484.5$. So, the new fatigue life evaluation model for FV520B-I in VHCF regime with $R = 0.5$ is written as:

$$Sc^{0.91} N_f = 484.5 \tag{14}$$

### 4.2. Further Assumption: The P-Sc-N Curve

As discussed above, the new fatigue life evaluation model can reasonably and quantitatively describe the relationship between fatigue cracking and fatigue life in a general condition (that is, the condition of reliability $R = 0.5$). The fatigue life model with $R = 0.5$ can satisfy most of application requirements of general practical engineering, just as the commonly used *S-N* curve represents a *P-S-N* curve with a reliability of 0.5, which can be applied in most cases. But *P-S-N* curves with other reliability values are still required to produce a comprehensive reliability analysis for FV520B-I in the VHCF regime. Thus, it is also necessary to improve the application range of the new fatigue life evaluation model by establishing fatigue life evaluation models with different reliability values, especially for some critical situations like $R = 0.1$ and $R = 0.9$. Based on the above points and requirements, a new comprehensive assumption is proposed to describe the relationship between fatigue reliability, fatigue cracking and fatigue life (that is, the *P-Sc-N* curve). The proposed *P-Sc-N* curve can improve the accuracy of the fatigue reliability analysis by considering the influence of fatigue cracking on fatigue failure, and it is also beneficial to the design of procedures in re-manufacturing engineering under the material fatigue failure control. The logical relationship between fatigue reliability, fatigue crack and fatigue life can be expressed as Figure 11:

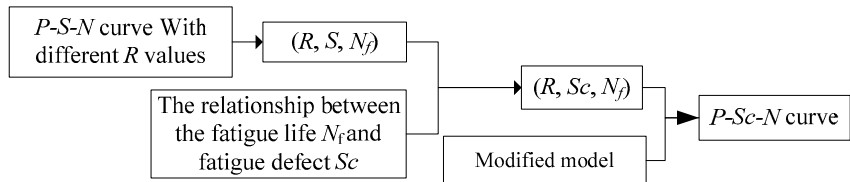

**Figure 11.** The logical relationship between fatigue reliability, fatigue crack and fatigue life.

As displayed in Equation (14), the new fatigue life evaluation model for FV520B-I is the *P-Sc-N* curve with $R = 0.5$, and *P-Sc-N* curves in critical conditions with $R = 0.9$ and $R = 0.1$ will be further developed to enrich the *P-Sc-N* curves of FV520B-I in the VHCF regime.

Referring to the derivation process of the fatigue life evaluation model with $R = 0.5$(Equation (14)), the nonlinear fitting algorithm was also adopted to identify the unknown model parameters. Then, the new *P-Sc-N* model of FV520B-I in VHCF regime with the reliability of $R = 0.9$ and 0.1 could

be specifically developed. For situations with $R = 0.9$, 0.5 and 0.1, the quantitative expressions between fatigue reliability, fatigue life and fatigue cracking for FV520B-I in the VHCF regime are written as:

$$\begin{cases} R = 0.9 : Sc^{0.68}N_f = 232.1 \\ R = 0.5 : Sc^{0.91}N_f = 484.5 \\ R = 0.1 : Sc^{1.28}N_f = 1074.1 \end{cases} \tag{15}$$

The new fatigue life evaluation models, with the consideration of fatigue reliability and fatigue cracking as expressed in Equation (15), also present *P-Sc-N* curve models of FV520B-I with three different fatigue reliabilities. Based on Equation (15), the *P-Sc-N* curves of FV520B-I with $R = 0.1$, 0.5 and 0.9 can be drawn as shown in Figure 12.

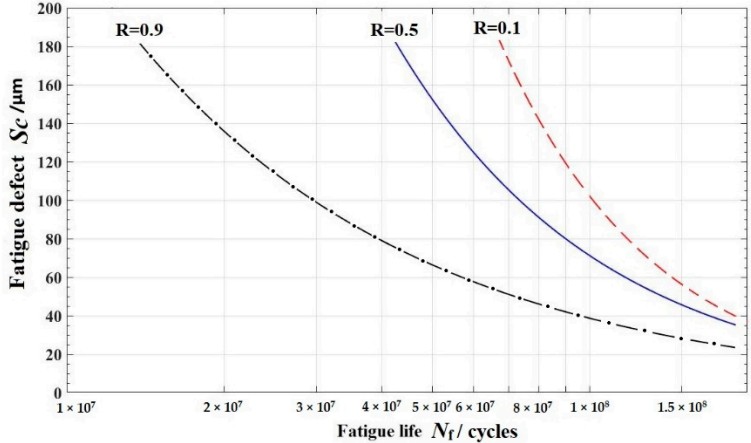

**Figure 12.** The *P-Sc-N* curve of FV520B-I with $R = 0.1$, 0.5 and 0.9.

### 4.3. Discussions

Figure 12 indicates some observations. First, both the fatigue life ($N_f$) and the corresponding fatigue reliability ($R$) can be determined based on the *P-Sc-N* curves with a certain fatigue crack value. The fatigue reliability analysis of FV520B-I in the VHCF regime is simplified to some extent.

Second, the *P-Sc-N* curve becomes more gradual with the increase of $R$ from 0.1 to 0.5, and this continues to 0.9. A small change in fatigue cracking will result in a large variation in fatigue life. This implies that fatigue life is more sensitive to the change in fatigue cracking when fatigue reliability is high.

Third, with an increase in fatigue life, the fatigue reliability appears to be more sensitive to the change in fatigue cracking. A small decrease in fatigue cracking will realize an improvement in reliability. For example, in the case of a $5 \times 10^7$-cycle fatigue life, fatigue reliability based on fatigue cracking is improved from 0.5 to 0.9 by a big decrease in fatigue cracking from about 150 μm to about 70 μm (the difference value is around 80 μm). However, in the case of a $1 \times 10^8$-cycle fatigue life, the fatigue reliability based on fatigue cracking is improved from 0.5 to 0.9 by a relatively small decrease in fatigue cracking, about 30 μm.

Fourth, in the case of the certain fatigue life, decreasing the fatigue crack is an effective and direct method to improve the fatigue reliability of FV520B-I in VHCF regime.

More than that, the proposition of the *P-Sc-N* curve is also useful to the decision-making process in remanufacturing based on the fatigue crack message. Remanufacturing is a kind of high-tech repair and transformation industry for the mechanical component with defect or to be scrapped parts. Based on the failure analysis, life assessment and other analysis, remanufacturing engineering design is carried out, and a series of related advanced manufacturing technologies are adopted to make the quality of remanufactured products reach or exceed new products. In remanufacturing applications, if the fatigue crack can satisfy the specification requirement, that is the existing fatigue crack *Sc* is smaller

than the fatigue crack critical threshold of remanufacturing *Ac*, the object will be re-manufactured. And the critical situation for remanufacturing is that *Sc* = *Ac* [1]. Then, different combinations of reliability and fatigue life (*R*,*N*f) in the critical situation for remanufacturing are also determined by the *P-Sc-N* curves. At this time, the fatigue reliability also presents the reliability that the fatigue crack and fatigue life satisfy the requirement of remanufacturing. The *P-Sc-N* curve of FV520B-I can also provide some useful information to the decision-making of remanufacturing judgment. The result can be optimized based on the *P-Sc-N* curve, by analyzing the changing relationship between the fatigue reliability and fatigue life under the certain fatigue crack, which provide a strong theoretical connection between material science and re-manufacturing practice.

## 5. Conclusions

A few conclusions can be drawn from this study:

(1)   The ultrasonic fatigue test for FV520B-I was performed. Obvious evidence of very-high cycle fatigue failure in FV520B-I, including "fish-eye" and GBF regions, were detected on the fracture surface; these are constant with the reported results of the very-high cycle fatigue study.

(2)   A three-parameter model for FV520B-I was achieved. The unknown parameters related to FV520B-I were obtained, and the fatigue reliability of FV520B-I in the VHCF regime was modeled. The corresponding *P-S-N* curves with different fatigue reliability values were drawn, respectively. The determination of the three-parameter model and the *P-S-N* curve for FV520B-I make up for the lack of a fatigue reliability study of FV520B-I in the VHCF regime.

(3)   A new fatigue evaluation model for FV520B-I, considering fatigue reliability and fatigue crack, was developed based on the *P-S-N* curves. This new model was verified using the test data and the errors were in an acceptable range.

(4)   A new relationship curve between fatigue reliability, fatigue crack and fatigue life *N* for FV520B-I in the VHCF regime was proposed: the *P-Sc-N* curve. The *P-Sc-N* curve model with different reliability values was developed with the combined application of the classic formula and test data. The corresponding *P-S-N* curves with different fatigue reliability values were drawn, respectively.

(5)   Using the *P-Sc-N* curve for FV520B-I, it was observed that there is a negative correlation between fatigue cracking and the reliability for FV520B-I. The new *P-Sc-N* curve is also useful for the decision-making process in re-manufacturing, based on the fatigue cracking information.

However, this research on fatigue reliability based on fatigue crack, is still a framework for potential research, and needs further investigation and deeper understanding. Specifically: (i) the rationality of the *P-Sc-N* curve has to be tested by a comprehensive application of fatigue experimentation and in actual engineering cases; (ii) more fatigue testing of FV520B-I is still required to modify the new model and the unknown model parameters, to improve the accuracy of the fatigue reliability analysis. The concerns discussed in this paper are obviously open to future research investigation and study.

**Author Contributions:** Conceptualization, J.W. (Jinlong Wang); methodology, J.W. (Jinlong Wang) and Y.Y.; software, Y.Y.; investigation, J.Y. and F.D.; resources, J.W. (Jingshi Wang); writing—original draft preparation, J.W. (Jinlong Wang); writing—review and editing, J.W. (Jinlong Wang), J.Y. and Y.Z. All authors have read and agreed to the published version of the manuscript.

**Funding:** This research was funded by the National Natural Science Foundation of China, grant number No. 51875082; National Defense Basic Scientific Research program of China, No.2019-JCJQ-JJ-547.

**Acknowledgments:** Thanks to Zhang Yuanliang, Ding Mingchao in Dalian University of Technology for their technical supports in this research work. And the authors have no conflict of interest to declare.

**Conflicts of Interest:** The authors declare no conflict of interest.

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
