# Peer review of "Fatigue Life Evaluation Considering Fatigue Reliability and Fatigue Crack for FV520B-I in VHCF Regime Based on Fracture Mechanics"

_metals, doi:10.3390/met10030371_

Round 1
Reviewer 1 Report
General comments
As the authors surely know, usually two fatigue mechanisms are present when testing at VHCF. The shorter ones starting at the surface (usually at higher stress levels, at the HCF regime), and the longer ones) starting at fish eyes (usually happening at lower stress levels at the VHCF regime. Did the authors not found fatigue failures encompassing both mechanisms?, i.e. both the ones originated at the specimen surface and those at fish eyes? In this case, are all these results assessed together? In the probable case that both kind of results are present, we are concerned with a particular problem of assessment: that of “confounded data”, whose treatment is not trivial.
The proposal of a log-normal distribution can be accepted to fit fatigue results though there are reasons to justify the use of extreme value distributions (Weibull, Gumbel, Fréchet), as for instance the interest of admitting only very low probabilities of failure in practical design. In fact, even 10% failure ratio would be considered excessive. But the main question is how to relate distributions each other resulting at the different stress amplitudes. The authors assume a linear relation in log-log scale, see Eq. (6), that is not supported by experimental data, if the S-N field experimental program would be continued to higher number of cycles (see [1]).
Indeed the authors’ proposal enhances the conventional two-parameter Basquin model including the asymptotic fatigue limit S, and incorporating a probabilistic analysis to the S-N field model and both complements are acknowledged as interesting ideas, but other limitations of the Basquin model, as its linearity on log-log scale remains. This linearity implies arbitrary coupling between the log-normal distributions resulting for the different stress amplitudes. Note that the proposed model is factually not a three parameter model, as the authors claim, but a five parameter model, where to the three-parameter corresponding to the basic “carrier model” defining the general trend of the S-N field, the two parameters of the log-normal distribution should be added.
The authors should admit that more advanced probabilistic fatigue models were already proposed by Freudenthal-Gumbel [2,3], Bolotin [4,5] and Castillo-Canteli [6] using aWeibull instead log-normal distribution. These models, contrary to the authors version, are soundly justified on the basis of statistical or micromechanical conditions. The free use program, ProFatigue, is available and prove to be able to face with VHCF results [7,8].
It is verified that the test program plan was highly ineffective, therefore, inadequate. With much less results at each of the amplitudes but suitably distributed along the S-N field, the authors would obtain more reliable information about an S-N field even extending it to higher number of cycles in further domain of VHCF. Lesser amplitudes had to be applied to confirm the trend of the S-N field to infinite lives and thus to estimate more reliably the fatigue limit. It can be accepted that for the short range of stress amplitudes considered, i.e. between 500 to 600 MPa, no notably difference could be observed in the S-N field assessment using an advanced fatigue model, as the ones mentioned above, instead of the proposed one. But the difference would be noticed when predicting fatigue lifetimes by extrapolating beyond the scope of the present fatigue program. Thus, according to the above comments, the present model is too conservative leading to anti-economic fatigue life prediction beyond 10^7 cycles which impedes a reasonably conservative (but not “too conservative”) design of components made of FV520B-I steel. Finally, the attempt to relate the fatigue results of the S-N field with the probabilistic fatigue field as a function of the initial crack growth is interesting, though crack growth rate curves could be recommendable for a reliable estimation of the Delta_K_th value.
In spite of the clear advantages of the Weibull fatigue models mentioned above, the proposed model could be accepted for publication in order to allow the interesting data presented about fatigue properties of FV520B-I steel. Nevertheless, the model proposed represents a confusing example for other researchers, which could infer that so many fatigue data are required for achieving a reliable fatigue assessment. In this way, this paper represents a non-recommendable pattern to be avoided in future fatigue programs. Consequently, mentioning of the alternative probabilistic S-N models would be inescapable.
The whole text must be revised by a native English speaker. English wording is often repetitive and poor. For instance, the word FV520B-I steel is unnecessarily used 6 times in the Abstract, 12 times in the conclusions and 27 times only in Section 1.
Under these conditions I proposed the work to be published under major revision giving answer to the questions posed.
[1] NIMS Fatigue Data Sheet No. 98, Data sheet on giga-cycle fatigue properties of Ti-6Al-4V (1100 MPa class) titanium alloy, National Institute for Materials Science, Tokyo, 2005.
[2] Freudenthal A.M., Gumbel E.J., On the statistical interpretation of fatigue tests, Proc. Roy. Soc., A216 (309),1953.
[3] Freudenthal A.M., Gumbel E.J., Physical and statistical aspects in fatigue, Advances in Applied Mechanics, Ed. H.J. Dryden & Th. Von Kármán, Academic Press, Vol. IV, 116-160, N.Y., 1956.
[4] Bolotin V.V., Statistical methods in structural mechanics, Holden-Day, S.F., 1969.
[5] Bolotin V.V., Wahrscheinlichkeitsmethoden zur Berechnung von Konstruktionen, Verlag für Bauwesen, Berlin, 1981.
[6] Castillo E., Fernández-Canteli A., A unified statistical methodology for modeling fatigue damage, Springer, 2009.
[7] Fernández Canteli A. et al., ProFatigue: A Software Program for Probabilistic Assessment of Experimental Fatigue Data Sets, Procedia Engineering, 74, 236-241, 201.
[8] M. Muniz-Calvente, A. Fernández-Canteli, B. Pyttel, E. Castillo, Probabilistic assessment of VHCF data as pertaining to concurrent populations using a Weibull regression model, Fat. Fract. of Eng. Mat. Struct., 40,1772-1782,2017.
[9] Stanzl-Tschegg S.E., When do small fatigue cracks propagate and when are they arrested?, Corros Review https://doi.org/10.1515/corrrev-2019-0023.
Particular comments
Page 2, lines 67-68: If the parameters are case oriented and not consistently referred as material properties irrespective of the specific application implied, one could suspect that the model is inadequate.
Page 6, line 178: The text is confusing: The stress range is two times the stress amplitude of the test.
Page 10, Fig. 8: The results lying outside the percentile curve R=0.1 of the PSN field may point out: a) that those results may not belong to the fish eye type but to other fatigue mechanism, which speculatively should be that initiated at the material surface or b) the proposed model is not adequate (as could be suggested by comparing it with the Weibull models already mentioned) since authors model implies linear fitting at a log-log scale, what contradicts VHCF results for very high number of cycles.
Page 10, lines 285 and 294: Replace “Figure9” by “Figure 8”.
Page 10, lines 286-302: The paragraph consists of obvious and elementary comments that are well-known to any fatigue researcher. In particular, the sentence stating that “stress amplitude is not equal to the mean stress, it is affected by the size of the fatigue crack” is not consistent: Possibly its means that “the influence of the stress amplitude depends on the mean stress and, particularly, on the fatigue crack size”.
Suggestions
Rewrite Abstract avoiding the reiterative mentioning of “FV520B-I steel”.
Page 1, line 11: I suggest to replace “Reliability and crack for FV520B-I” by “reliability and crack growth mechanism of FV520B-I”. See also page 2, line 72.
Page 1, lines 33-38: The text is redundant: “FV520B-I” is used unnecessarily 5 times in this paragraph, “reliability” 4 times and “VHCF” 3 times.
Page 1, lines 39-50: “Reliability” is repeated 9 times.
Page 2, line 53: replace “stress characteristics” by “stress distribution”.
Page 2 line 54: replace “made a fatigue analysis of FV520B-I impeller” by “performed its fatigue analysis”
Page 5, lines 143-144 and 146-147: Text is reiterative
Page 6, lines 184-186. The text is reiterative.
Page 15, Conclusions:
Conclusion 1: The sentence “the very-428 high cycle fatigue failure of FV520B-I is caused by the fatigue crack” is obvious and unspecific, I suggest to relate “fatigue crack” with “fish eye” in this sentence.
Conclusion 5: “…it can be observed that decreasing the fatigue crack is a direct and effective way to improve the fatigue reliability and life at the same time in engineering application for FV520B-I.” is an obvious statement. The question is to explain the solution for reduce effectively the fatigue cracks in components made of FV520B-I steel.
References: Some relevant bibliography, as that provided by S. Stanzl-Tschegg (see [9]), would be recommended to consider in the Reference list. It is focused on this specific problem though not necessarily referred to FV520B-I steel.

Reviewer 2 Report
Dear authors,
congratulations for this idea of research. I must point out some ítems:
scientific views:
Table 1. Chemical composition. You should put in order chemical elements ( first C....last Fe ( bal.) etc.
Table 2. You should remove units from HV file
lines 106-108 it is necessary justify the aim of this particular termal treatment, quench and temper
Figure 2. It is necessary identify a) and b)
Figure 3. Attending comments of lines182-189 you should insert some lines to identify your study zones attending stress amplitude
Figure 4 and 5. it is necessary to put magnification into figure´s legend
I miss microstructural study of this steel since could provide a lot of information that justify both fracture observed types, GBF and "fish eye". I recommend you the paper Y. Hong et al. Int J Fatigue 89, 2016, pp.108-118. It is linked with VHCF also
Did you observe micropits onto fracture Surface? Justify
Define with clarity S0R.
Why you not apply Fuzzy Logic to this reliability factor/concept?
line 259 and so on: Take care Figure 9 ( it needs a space)
Please, clarify in deep the concept of remanufacturing and give the industrial and reallistic aplication of it in relation with the practical concern of your paper
Yes, your last remark is necessary (lines 448-453) : you are setting the first steps to develop a new model of fatigue life evaluation but needs tests to be sure of your affirmations.
Reviewer 3 Report
Comments on the paper
- Page 3 – in the literature there is the designation large R as stress ratio.
- Fig. 2 – Fig. 2a is difficult to read, you may describe it? in the caption Fig. 2b suggest - The specimen and its dimensions.
- Figs 3, 7, 8, 9, 10, 12 – missing unit at Fatigue life - in cycles (hours)?
- Fig. 3 – the caption under the figure should be changed to “The test results for fatigue”.
- Fig. 4 – enter the markings a) and b) in the caption also.
- Fig. 5 – as above in Fig. 4.
- Page 6 – Eq. (1) if it is not original (of the authors), please provide literature.
- Page 7 – no explanation of symbols and markings like: P-S-N, and others.
- Page 10 – Can you cite literature confirming the results obtained in the discussion?
- It would be worthwhile to quote in the introduction also papers of: 1) Smaga M., Boemke A., Daniel T., Skorupski R., Sorich A., Beck T. Fatigue behavior of metastable austenitic stainless steels in LCF, HCF and VHCF regimes at ambient and elevated temperatures. Metals 2019, 9, 704, 2) Rozumek, D.; Marciniak, Z.; Lesiuk, G.; Correia, J.A.; de Jesus, A.M. Experimental and numerical investigation of mixed mode I+ II and I+ III fatigue crack growth in S355J0 steel. Int. J. Fatigue 2018, 113, 160-170.
Round 2
Reviewer 1 Report
Answer to authors’ points
General comments
Point 1: OK
Point 2: The rationality of Eq. (6) is put under question. No one could demonstrate that Eq. (6) can be used to fir adequately fatigue experimental results in HCF and much less in VHCF near the fatigue limit. The new citation is only an example (1988) of a fatigue program in which the HCF region is not investigated for long lives close to fatigue limit. Figs. 1 and 2 in the publication mentioned by the Authors evidence the inadequate planning of the fatigue results. They are namely located in the seemingly linear region of the S-N field, but no extrapolation can be rationally applied for fitting results in the HCF region. On the contrary, I can provide the authors hundreds of examples in the opposite sense . For instance, look at the many contra-examples in E. Haibach, Betribsfestigkeit, VDI Verlag, 1989, pages 32, 33, 36, 38, etc.
Point 3: The problem of the P-S-N models is just that explained by the Authors: The use of two steps in assessing the probabilistic S-N field. The two steps are independently derived and therefore disconnected and further artificially enforced to be coupled at tehh S-N curve for 50% probability of failure. As a result, this way of tackling the S-N field, does not fulfil the necessary statistical conditions a valid fatigue model must accomplish, see
[1] Freudenthal A.M., Gumbel E.J., On the statistical interpretation of fatigue tests, Proc. Roy. Soc., A216 (309),1953.
[2] Freudenthal A.M., Gumbel E.J., Physical and statistical aspects in fatigue, Advances in Applied Mechanics, Ed. H.J. Dryden & Th. Von Kármán, Academic Press, Vol. IV, 116-160, N.Y., 1956.
[3] Bolotin V.V., Statistical methods in structural mechanics, Holden-Day, S.F., 1969.
[4] Bolotin V.V., Wahrscheinlichkeitsmethoden zur Berechnung von Konstruktionen, Verlag für Bauwesen, Berlin, 1981.
[5] Castillo E., Fernández-Canteli A., A unified statistical methodology for modeling fatigue damage, Springer, 2009.
Point 4: I appreciate Authors’ experimental work for carrying out such an extensive experimental program for FV520B-I), which is their most valuable contribution for practical design. I only regret that they have ignored other ways of planning the experimental program for determining the S-N field, with less specimen number and higher reliability.
Point 5: Surely, the accuracy of the analysis can be improved by increasing the number of test data but this is not necessarily the best way to achieve it if neither an adequate probabilistic fatigue model nor an adequate planning, according to the probabilistic model, are available.
Point 6: OK
Particular comments
Pont 1: The assessment for different materials and the experimental planning can be ruled by the same fatigue model, if this is a valid one, even if the model parameters could be diverse.
Point 2: OK
Point 3: If the experimental results do not fall within the expected range as provided by the theoretical model, this is not correct and must be rejected. Otherwise you should provide the expected scatter in advance.
Remaining points: OK
Finally I insist that : “the model proposed represents a confusing example for other researchers, which could infer that so many fatigue data are required for achieving a reliable fatigue assessment. In this way, this paper represents a non-recommendable pattern to be avoided in future fatigue programs. Consequently, mentioning of the alternative probabilistic S-N models would be inescapable.”
Reviewer 2 Report
Dear authors,
you have really improve the paper. Congratulations, Only a minor comment: it would be nice that include photo magnification in legend on Figures 4 and 5. For example:
Figure 4 a)…….-.x80; b)……...x300
and so on.
Best Greetings,
Reviewer 3 Report
The authors took into account all comments of the Reviewer. After considering the comments, the work is more understandable to the reader. Based on the above mentioned, I recommend publishing the article in the journal Metals.
